# A Look at Plant-Growth-Promoting Bacteria

**DOI:** 10.3390/plants12081668

**Published:** 2023-04-17

**Authors:** Lorena Jacqueline Gómez-Godínez, José Luis Aguirre-Noyola, Esperanza Martínez-Romero, Ramón Ignacio Arteaga-Garibay, Javier Ireta-Moreno, José Martín Ruvalcaba-Gómez

**Affiliations:** 1Centro Nacional de Recursos Genéticos, Instituto Nacional de Investigaciones Forestales, Agrícolas y Pecuarias, Tepatitlán de Morelos 47600, Jalisco, Mexico; 2Centro de Ciencias Genómicas, Universidad Nacional Autónoma de México, Av. Universidad s/n, Cuernavaca 62210, Morelos, Mexico; 3Centro de Investigación Regional Pacífico Centro, Centro Altos Jalisco, Instituto Nacional de Investigaciones Forestales, Agrícolas y Pecuarias, Tepatitlán de Morelos 2470, Jalisco, Mexico

**Keywords:** microorganisms, biofertilizer, agriculture

## Abstract

Bacteria have been used to increase crop yields. For their application on crops, bacteria are provided in inoculant formulations that are continuously changing, with liquid- and solid-based products. Bacteria for inoculants are mainly selected from natural isolates. In nature, microorganisms that favor plants exhibit various strategies to succeed and prevail in the rhizosphere, such as biological nitrogen fixation, phosphorus solubilization, and siderophore production. On the other hand, plants have strategies to maintain beneficial microorganisms, such as the exudation of chemoattractanst for specific microorganisms and signaling pathways that regulate plant–bacteria interactions. Transcriptomic approaches are helpful in attempting to elucidate plant–microorganism interactions. Here, we present a review of these issues.

## 1. Plant-Growth-Promoting Rhizobacterias (PGPRs)

The need to produce food in large quantities driven the implementation of several production techniques, increased inputs, more productive varieties, and an increasing acceleration in crop cycles. Intensive agricultural use of these production technologies has caused an excessive use of chemical fertilizers and therefore a damage to the environment [1,2]. Alternatively, microorganisms from soil and plants may be used, especially rhizobacteria, which have lived with plants and have evolved in parallel with them. Plants provide a habitat for microorganisms due to the root exudation of organic compounds that are necessary for microbial metabolism [3,4,5]. Such microorganisms can live inside or outside their host, abundantly colonizing the plants’ rhizosphere or root surface [3,6].

Over time, the use of microbial inoculants or biofertilizers in agriculture has increased. Biofertilizers are products formulated from living microorganisms that are capable of promoting growth in plants of agricultural interest. The first microorganisms used as inoculants were “rhizobia”, diazotrophic bacteria capable of colonizing the rhizosphere and establishing nodules in the roots of their host legume. The first commercial biofertilizer was the so-called “Nitragin”, formulated from rhizobia; Nobbe and Hiltner (1896) patented this biofertilizer, which used gelatin as a nutrient medium for microorganisms. Later, at the beginning of the 20th century, formulations were based on solid carriers such as soil and peat [7,8]. Years later, liquid vehicles were formulated to sell biofertilizers based on additives such as polyvinylpyrrolidone (PVP), carboxymethylcellulose (CMC), arabic gum, sodium alginate, glycerol, and cell protectors to improve growth performance. Additives promote cell survival in storage and after their application to seeds or soil [9,10,11].

Currently, many products on the market are composed of one or combinations of plant-growth-promoting rhizobacteria (PGPRs). Among the most commonly studied rhizobacteria are *Aminobacter*, *Arthrobacter*, *Azoarcus*, *Azospirillum*, *Azotobacter*, *Bacillus*, *Bradyrhizobium*, *Brevundimonas*, *Burkholderia*, *Clostridium*, *Delftia*, *Enterobacter*, *Gluconoacetobacter*, *Herbaspirillum*, *Klebsiella*, *Paenibacillus*, *Rhiseudobacillus*, *Serratia*, *Sphingomonas*, and *Xanthomonas*. However, the most used bacteria for the formulation of biofertilizers are *Azospirillum*, *Bacillus*, and *Pseudomonas*, in addition to some endo-ectomycorrhizal fungi such as *Rhizophagus irregularis*, *Gigaspora margarita*, *Pisolithus tinctorius*, and *Scleroderma citrinum* [1,12,13,14]. Table 1 lists microorganisms with beneficial effects in different crops.

## 2. Direct and Indirect Mechanisms of Microorganisms in Plants

PGPRs can act in two ways to improve the fitness of its hosts, either directly or indirectly. Among the direct mechanisms are those that promote plant growth. Within these, we find the production of hormones, such as auxins, cytokinins, and gibberellins, as well as nitrogen fixation and phosphorus solubilization. The indirect mechanisms are related to inhibiting the function of one or more plant-pathogenic organisms [25]. These main mechanisms are the production of antibiotics, enzymes that degrade the cell wall and antioxidants, the inhibition of the pathogen quorum, induced systemic resistance, and iron sequestration by bacterial siderophores. The reduction in ethylene levels by the enzyme 1-aminocyclopropane-1-carboxylate (ACC) deaminase is also classified as a direct mechanism of promotion (Figure 1) [26,27,28].

The production of auxins is one of the most frequently reported for the promotion of plant growth: it has been found that 80% of microorganisms colonizing the rhizosphere can produce this metabolite. For decades, an increase in the number of root hairs and lateral roots and a shortening of root length have been reported when plants are inoculated with PGPRs. These root morphological changes have been attributed to bacterial auxin production, and studies with plant mutants altered in IAA production confirmed its role [29]. The plant-growth-promoting effect of bacteria in which auxin is involved is known as phytostimulation. Root nodules have been reported to contain more auxin than non-nodulated roots [30]. Conversely, gibberellins include a large group of tetracyclic diterpenoid carboxylic acids with C20 or C19 carbon skeletons [31]. Gibberellins are known to stimulate growth and activate growth processes, such as stem elongation, seed germination, flowering, and fruit setting [32], and increase the rate of photosynthesis and the content of chlorophyll [33,34]. Within the genera of PGPRs that can produce gibberellins are *Bacillus* spp., *Achromobacter xylosoxidans*, *Gluconobacter diazotrophicus*, *Acinetobacter calcoaceticus*, *Rhizobium*, *Azotobacter* spp., *Herbaspirillum seropedicae*, *Enterococcus faecium*, *Pseudomonas* spp., *Promicromonospora* spp., and *Azospirillum* spp. [31,35]. Other phytohormones produced by microbes are cytokinins. They control cell differentiation in meristematic tissues and regulate apical dominance, root elongation, seed germination, flower and fruit development, and plant–pathogen interactions [36,37]. The PGPRs that produce them are *Bacillus subtilis, Methylobacterium*, and *Sinorhizobium meliloti* [38,39].

Nitrogen is one of the most important macronutrients for plant growth. The abundance of nitrogen in the atmosphere is approximately 78%, and plants cannot assimilate it. Many PGPRs have been identified that can carry out biological nitrogen fixation freely or under symbiotic associations with legumes. Some examples of symbiotic nitrogen fixers are *Rhizobium*, *Sinorhizobium*, *Azorhizobium*, *Allorhizobium*, *Mesorhizobium*, *Bradyrhizobium*, *Burkholderia*, and *Herbaspirillum* [1,40]. Cyanobacteria (e.g., *Nostoc* and *Anabaena*), *Azoarcus, Azospirillum*, *Azotobacter*, *Enterobacter*, *Gluconacetobacter*, *Klebsiella*, *Herbaspirillum*, and *Pseudomonas* can be free-living nitrogen fixers or establish associations with plants as endophytes [41,42,43]. The inoculation of nitrogen-fixing microorganisms in seeds, seedlings, roots, or soil stimulates plant growth, improves soil quality, and maintains the nitrogen level in the soil [44].

Phosphorus is another essential plant macronutrient since it functions in different metabolic processes, such as photosynthesis. It is found in molecules such as ATP and nucleic acids and is involved in signal transduction. However, there is a drawback since more than 90% of the phosphorus in the soil is insoluble, immobilized, or precipitated, making it difficult for plants to absorb it. In the soil, some bacteria solubilize inorganic phosphorus; these bacteria are called phosphate-solubilizing bacteria [45]. They convert insoluble organic and inorganic phosphate into a bioavailable form for plants. Some PGPRs, such as *Arthrobacter*, *Bacillus*, *Burkholderia*, *Enterobacter*, *Microbacterium*, *Pseudomonas*, *Rhizobium*, *Mesorhizobium*, *Flavobacterium*, *Rhodococcus*, and *Serratia*, are listed in that category [46]. In addition, these phosphate solubilizers also stimulate plant growth [47]. Another crucial nutrient required by plants is iron (Fe). However, similar to the previous nutrients, it is also unavailable to plants since it is insoluble in Fe^3+^, associated with hydroxides and oxyhydroxides [48]. Some PGPRs can secrete siderophores in soil: phenolates, catecholates, hydroxamates, carboxylates, or mixed types. Siderophores are small peptide molecules that bind Fe^3+^ and make it available to cells [49]. Some siderophores also show an affinity for Pb, Cd, Zn, Cu, Co, Mo, and even for As. PGPRs with this ability are *Pseudomonas*, *Bacillus*, *Rhizobium*, *Azotobacter*, *Enterobacter*, and *Serratia* [50].

Among the indirect mechanisms, the most important is the production of antibiotics such as surfactin, fengycin, rhamnolipids, phenazine-1-carboxylic acid (PCA), pyrrolnitrine, butyrolactones, zwittermicin A, aerugin, azomycin, cepafungins, kanosamine, and karalicin [49]. They can act as antifungal, antibacterial, anthelmintic, and antiviral agents. Bacterial peptides can inhibit phytopathogenic fungi such as *Rhizoctonia*, *Fusarium*, *Pythium*, *Alternaria*, *Phytophthora*, and *Botrytis*. PGPRs belonging to the genera of *Bacillus*, *Pseudomonas*, and *Streptomyces* have been exploited or control plant diseases in many economically important crops [43,51,52].

Some organisms can produce enzymes that can degrade the fungal cell wall of some phytopathogens. Among the enzymes with this ability are chitinase, β-1,3-glucanase, protease, and lipase, which are responsible for the degradation of the components of fungi cell walls [53,54,55]. On the other hand, when plants are subjected to different types of stress they produce ROS reactive oxygen species, which are related to oxidative cell damage. Therefore, some microorganisms can produce an antioxidant defense system for plants; an example of these mechanisms is the production of antioxidant enzymes such as the catalase (CAT), superoxide dismutase (SOD), and peroxide dismutase (POD) produced by some genera of PGPRs such as *Pseudomonas fluorescens*, *Bacillus amyloliquefaciens*, and *Bacillus licheniformis* [56,57,58]. Bacteria can also communicate through signaling molecules called N-acyl homoserine lactones (AHL), which may also serve to detect environmental changes and bacterial population density. Some plant-pathogenic microorganisms have this ability to communicate through molecules and thus become more virulent [59]. Disrupting quorum sensing through PGPRs, which can produce some enzymes such as lactonase, is a strategy to avoid communication between pathogens, diminishing virulence and inhibiting their growth in the plant [60]. Among the genera with this capacity are *Bacillus*, *Agrobacterium*, *Rhodococcus*, *Streptomyces*, *Arthrobacter*, *Pseudomonas*, and *Klebsiella* [61].

Induced systemic resistance (ISR) is a plant defense mechanism triggered by pathogen infection, injuries, or root colonization. Induced systemic resistance (ISR) can be elicited by PGPRs through their cell wall components or metabolites [25]. Some PGPRs, such as *Bacillus subtilis* and *Pseudomonas* sp., induce ISR and produce antimicrobial peptides that control infection through phytopathogens (e.g., *Rhizoctonia*, *Fusarium*, *Pythium*, *Alternaria*, *Ralstonia*, *Phytophthora*, and *Botrytis*) [62,63].

Plants can select their microbiota by releasing root exudates. Microbes are attracted by chemotaxis to the rhizosphere by secondary metabolites, low molecular weight (amino acids, organic acids, and sugars), and high molecular weight compounds (polysaccharides, mucilage proteins, and vitamins) [64]. L-malic acid and other nutrients play a role in the host recognition by *Bacillus subtilis* FB17 but not of other *Bacillus* sp. [65]. The microbiota in the rhizosphere of plants can include algae, archaea, arthropods, bacteria, fungi, nematodes, protozoa, or viruses [64]. Microorganisms have developed strategies to grow and persist in the rhizosphere. They can degrade antimicrobial metabolites secreted by plants (e.g., phytoalexins, flavonoids, and alkaloids), avoiding their toxic effects. The molecular mechanisms governing these interactions are reviewed below from a transcriptomic point of view.

Although the rhizosphere has been the main source of growth-promoting bacteria, the phyllosphere is also a reservoir of microorganisms with outstanding metabolic potential. The term phyllosphere refers to the aerial parts of plants, including stems, leaves, fruits and reproductive structures [66]. Leaves comprise the major portion of the phyllosphere; there are approximately 10^6^–10^8^ bacterial cells/cm^2^ in each leaf [67]. Phyllosphere microbiota can have an endophytic origin, or they may come from the air, rain or irrigation water, vectors, or soil dust. However, not all microbes resist the exposure to UV rays, nutrient starvation, and environmental temperature and humidity fluctuations [68]. They must also successfully attach to leaf cuticle, a useful characteristic when bacteria are used as foliar-applied bioinoculants. From there, they release siderophores, volatile organic compounds, and antimicrobial metabolites [69]. Moreover, microbes in the phyllosphere can trigger systemic responses in plants that increase the production and accumulation of phytoalexins, alkaloids, and glucanases that avoid phytopathogen invasion [68]. Other roles of these microbes involve the control of flowering, seed and fruit development, protection against contaminants and pesticides, improvement of crop yields, and carbon and nitrogen cycling [70]. *Sphingomonas*, *Streptomyces*, *Pseudomonas*, *Methylobacterium*, and *Bacillus* are commonly found as phyllosphere residents in maize, rice, soybean, sugarcane, and fruit trees [69]. Some of them achieve colonization of plants through stomata, lenticels, and hydathodes and are distributed to other tissues via the xylem and phloem systems [71]. A combination of rhizosphere and phyllosphere bacteria would be a novel approach to producing biofertilizers that meet key aspects of plant nutrition.

## 3. Plant Strategies to Select Beneficial Microorganisms

In both natural and agricultural ecosystems, beneficial and pathogenic microorganisms, mainly bacteria and fungi, colonize plants [72]. The plant–microbe interaction is a complex, dynamic, and ongoing process as old as plant colonization on Earth. The association of the multicellular host, in this case, the plant, with its associated microbiota generates a functional entity called a “holobiont” [73,74,75]. However, in plants, this concept has been questioned for its evolutionary implications because plants associate only transiently with many microbes, in many cases with redundant functions [76]. Plants are continually in contact with many different individual microorganisms, including bacteria, fungi, viruses, and protists. The establishment of communities of microorganisms in plants is not random: some conditions control it, for example, the type of soil, the genotype of the host, the stage of development of the plant, and the plant’s organ [77,78,79]. Although most microbes encountered by plants are naturally commensal, a small but significant portion of them will go on to form pathogenic or mutualistic symbioses with the plant.

So far, numerous studies have revealed plant–microbe interaction processes such as how plants respond to microbial colonization, including pathogens. However, there are still questions on how plants differentiate between beneficial and pathogenic microbes or between different species of pathogens or how gene regulatory networks and signal transduction pathways control these processes [80]. The best-studied symbiotic interactions occur between plants and arbuscular mycorrhizal fungi (AMFs) and between legumes and nitrogen-fixing rhizobia. This interaction is achieved thanks to chemical communication between microorganisms and plant roots. Plant roots release strigolactones and AMFs produce Myc factors, followed by a series of cellular changes that begin in the roots and lead to the formation of a penetration apparatus, a channel through which the fungal hyphae can colonize. In legumes, roots secrete flavonoids detected by rhizobia, which will release Nod factors in response. Nod factors induce the formation of an infection thread in the roots, which serves as an entry point for rhizobia and initiates the development of nitrogen-fixing nodules [81]. Microbes on or in plant tissue produce many different signals, including volatile organic compounds, hormones, hormone mimics, and carbohydrate- and protein-based signals, for symbiotic interaction and in response to defense [82].

Many genes necessary for the interactions between plants and bacteria and plants and fungi have been discovered and described including. The common symbiotic signaling pathway (CSSP) pathway. This pathway includes kinases and co-receptor proteins that sense the presence of rhizobial bacteria or AM fungi and a series of signaling proteins, calcium/calmodulin, which induce gene expression necessary for establishing mutualism. There is evidence that certain CSSP elements are sequestered during the colonization stage of plant tissues in plants. CSSP mutations in *Medicago truncatula* were affected inbeneficial and pathogenic interactions [83].

Plants recognize conserved molecules from microbes; these metabolites are also known as microbial- or pathogen-associated molecular patterns (MAMPs or PAMPs). Plants have evolved different pattern recognition receptors (PRRs) in the plasma membrane that bind MAMPs and PAMPs and control plant immune responses. In response to PAMPs, plants elicit a defense response termed PAMP-activated immunity (PTI) or basal resistance, the first level of defense that restricts infection by pathogens in most plant species [84].

The role of plant miRNAs in recent years has gained considerable interest since it is known that miRNAs are involved in signaling pathways associated with symbiotic interactions. miRNAs are small, endogenous non-coding RNA fragments (18 to 24 nucleotides) that affect the stability and translation of mRNAs and act as the primary post-transcriptional regulators of gene expression [85]. Different plant miRNAs have been implicated in modulating pathogenic microbial interactions with plants. miRNAs involved in pathogen resistance function as regulators of the NBS-LRR genes, which encode proteins characterized by a nucleotide binding site (NBS) and a leucine-rich repeat (LRR). For example, miR482 favors potato resistance by suppressing the NBS-LRR genes during *Verticillium dahliae* infection; in tomatoes, the decrease in the expression of miR482 and the increase in the expression of BS -LRR confer resistance to *V. dahliae* [86].

Nobori et al. (2018) evaluated 27 combinations of plant interaction and immunity between *Arabidopsis thaliana* and the common pathogen *Pseudomonas syringae*, identifying genes and specific bacterial processes of “immune response”. In *P. syringae*, it was possible to identify bacterial transcriptomic signatures influenced by plant immune activation and the overexpression of the sigma factor gene involved in iron regulation and how this partially decreases bacterial growth during plant immunity [87]. The analysis of flagellin 22 (flg22), which is encoded by hundreds of *Arabidopsis* symbionts genomes, has revealed that these peptides allow for evasion of the immune response. Therefore, plants can identify the type of bacterial flagellin and use this information to identify pathogens [88].

In general, plants may differentiat beneficial microorganisms from pathogens in different ways, such as recognizing different flagellar epitopes, receptor competition, and post-transcriptional regulation through miRNAs.

## 4. Mechanisms of Plant–Bacteria Interaction: A Transcriptomics View

Plant–microbe interactions involve chemical signaling in which microorganisms detect plant-derived metabolites and express genes associated with chemotaxis, nutrient metabolism, and the synthesis of plant growth regulators [89,90]. This inter-kingdom dialogue may not always be the same, depending on the host, the time of interaction, the chemical nature of metabolites, or the presence of other microorganisms or plants [91,92,93]. The goal of transcriptomics is to explore the complete expression of genomes in a defined condition; hence, it has been a valuable tool for deciphering the mechanisms involved in both sides of PGPRs’ associations with plants [94,95].

The composition of root exudates differs according to genotype, age, and nutritional conditions, but they generally consist of sugars, amino acids, organic acids, and secondary metabolites [96]. Trials of PGPRs under the influence of root exudates have made it possible to simulate the interactions that occur in the rhizosphere. For example, *Bacillus subtilis* OKB105 expressed genes for glucose, arabinose, and malic acid transport in response to rice exudates [97], whereas in contact with maize exudates, *B. amyloliquefaciens* SQR9 expressed genes for the metabolism of inositol, mannitol, alanine, glutamate, lysine, and aspartate [98] (Figure 2). Transcriptomes of *Rhizobium leguminosarum* and *R. phaseoli* interacting with different host plants showed the preference of rhizobia for the uptake of organic acids based on the overexpression of genes encoding specialized C4-dicarboxylate and tricarboxylate transport systems [99,100]. Conversely, the exposure of *Bradyrhizobium* to soybean exudates has been linked to the expression of genes involved in the IAA metabolism, suggesting that legumes may regulate the production of auxin by bacteria [101]. In addition, *Azospirillum brasilense* reaches high levels of expression of genes for auxin production and for biological nitrogen fixation when colonizing maize rhizoplane in the presence of other PGPRs, indicating that metabolites produced by one member of the rhizosphere community can affect the gene expression of others [93] (Figure 2).

In poor soils, the plants activate a “cry for help” system which attracts microorganisms with outstanding metabolic capabilities. Root exudates can activate genes of PGPRs involved in root attachment and colonization, including those encoding secretion systems, fimbriae, flagella, plant wall-degrading enzymes (cellulases, xylanases, pectinases, and endoglucanases), and quorum-sensing systems [102,103]. Some secretion systems act as a molecular syringe that releases their effectors into the plant cell, weakening the immune response and promoting bacterial entry [104]. When *Bradyrhizobium japonicum* USDA 110 and *Sinorhizobium fredii* HH103 sensed root flavonoids, they expressed a type III secretion system (T3SS) necessary to establish effective nodulation and nitrogen fixation with soybean (*Glycine max*) [105]. Likewise, the type VI secretion system (T6SS) of *Rhizobium etli* Mim1 is essential for symbiosis with common beans [106]. Secretion systems also play an important role in PGPRs for interaction with cereals. The nitrogen fixer *Azoarcus* sp. BH72 expressed genes encoding a T6SS and pili in response to rice exudates. Mutants in these genes are deficient in root invasion [107]. The genes encoding T3SS, endoglucanases, and polygalacturonases were activated in *R. phaseoli* in the presence of maize exudates [100]. In contrast, when this bacterium colonized the rhizoplane, it expressed a T6SS secretion system [108]. A fascinating dual transcriptomic/proteomic study to determine key genes for adaptation to an endophytic lifestyle revealed that *Nitrospirillum amazonense* expressed a type IV secretion system and RND efflux pumps for toxic compounds in sugarcane apoplast [109].

Root exudates are rich in nutrients and have phytoalexins, flavonoids, phenolic acids, and alkaloids with antimicrobial activity [92]. The abundance of these compounds in the rhizosphere shapes the assembly of microbial communities [110]. PGPRs carry mechanisms for detoxifying plant metabolites through degradation, neutralization, or excretion. Work on rhizosphere-associated *Pseudomonas*, including *P. fluorescens*, *P. brassicacearum*, *P. chlororaphis*, *P. protegens*, and *P. synxantha* showed an overexpression of genes for the cleavage of catechol, benzoate, and phenolic acids in interaction with the grass *Brachypodium distachyon* [111] and sugarbeet [112]. *Rhizobium etli* CFN42 harbors an efflux pump, RmrAB, that confers tolerance to naringenin, phaseollin, phaseollidin, coumaric acid, and salicylic acid [113] (Figure 2). In the phytopathogen *Pseudomonas syringae* pv. tomato DC3000, the MexAB-OprM efflux pump genes confer protection to flavonoids and phenolic compounds, providing it with a good fitness in the rhizosphere and an advantage in the colonization of tomato plants [114].

In addition to plant-growth-promoting capabilities, other PGPR mechanisms protect against phytopathogens and are induced by plants. *Brachypodium* root exudates impact the regulation of *Pseudomonas* gene expression for the production of the antifungals pyrrolnitrin, phenazine-1-carboxylic acid, and 2,4-diacetylphloroglucinol [111]. Expression of genes for the biosynthesis of bacillaene, difficidin, surfactin, macrolactin, and fengycin by *B. amyloliquefaciens* was detected in an interaction with maize [98,115]. At the same time, *Rhizobium* expresses genes for the production of the siderophore vicibactin under similar assays [100]. This spectrum of secondary metabolites is efficient for the biocontrol of plant pathogenic fungi such as *Fusarium oxysporum, F. verticiloides*, *Gibberella moniliformis*, *Bipolaris oryzae*, and *Colletotrichum gloeosporioides* [116,117,118].

The beneficial effects of bioinoculants on different crops have been widely described at the agronomic level [119,120]; however, there are many mysteries about the molecular mechanisms that lead to changes in plant physiology and what their microbial inducers are. A transcriptomic view during plant–microbe crosstalk proves that PGPRs can modify global gene expression in their hosts. Plants can distinguish whether a pathogenic or commensal microorganism colonizes them by detecting MAMPs. These include lipopolysaccharides, peptidoglycans, flagellin, and fungal chitooligosaccharides [121,122]. Hence, systemically --induced responses can be induced by PGPRs such as rhizobia [123,124], *Bacillus* [125], and *Pseudomonas* [126], as well as yeasts [127] and filamentous and mycorrhizal fungi [128]. A significantly greater number of differentially expressed genes have been observed in plants colonized by pathogenic bacteria than in commensal bacteria. When *A. thaliana* was inoculated with a beneficial bacterium (*Methylobacterium extorquens* PA1), transcriptomic responses that modulate antioxidant mechanisms and metal homeostasis were detected. However, infection with the pathogen *Sphingomonas melonis* Fr1 increased gene expression in phytoalexin biosynthesis [129]. Phytoalexins are antimicrobial against various fungi and bacteria; some have allelopathic effects. In another study, genes for abscisic acid biosynthesis and signaling in *A. thaliana* were upregulated in the presence of the pathogenic bacterium *Ralstonia solanacearum* [130]. *Abscisic acid* is a sesquiterpenoid that regulates adaptive responses to abiotic and biotic stresses [131].

Other types of responses have been detected during interaction with PGPR. Hardoim et al. (2020) [132] found that inoculation with *A. brasilense* sp245 induced the activation of genes of maize associated with apoplast metabolism, while *Herbaspirillum seropedicae* HRC54 induced the expression of genes associated with auxin-dependent signaling and antioxidant mechanisms and activated gibberellic-acid-regulated pathways in plants but repressed abscisic acid biosynthesis. In agreement, the interaction of *P. putida* with chickpea (*Cicer arietinum* L.) modulated the expression of miRNAs targeting genes associated with drought survival [133]. In tobacco plants, genes associated with systemic responses mediated by jasmonic acid and salicylic acid were activated in the presence of *Paenibacillus polymyxa* YC0136, including genes for the production of phenylpropanoids and laccases [101]. Jasmonic acid and salicylic acid are plant-growth regulators, but their role in triggering resistance and senescence responses has been extensively described [134].

Not all the interactions between plants and microorganisms are physical; volatile organic compounds (VOCs) also serve as a communication channel. The gases emitted by the actinobacterium *Microbacterium aurantiacum* GX14001 modified the tobacco transcriptome by positively regulating genes associated with phenylpropanoid, flavonoid, and gingerol biosynthesis [135]. Likewise, *B. subtilis* GB03 was shown to activate cytokinin pathways in *Arabidopsis thaliana* through its VOCs, mainly by 2,3-butanediol and 3-hydroxy-2-butanone [136]. *Pseudomonas pseudoalcaligenes* produces dimethyl disulfide, 2,3-butanediol, and 2-pentyl furan; these VOCs are responsible for increasing germination, modulating osmolyte concentration, and alleviating drought stress in maize [137]. Recently, new rhizobial VOCs, such as methylketone 2-tridecanone (2-TDC), showed the ability to regulate the growth of *Medicago truncatula* and sorghum (*Sorghum bicolor*) and to protect tomato (*Solanum lycopersicum*) against infection by *P. syringae* pv. tomato [138]. Knowing the microbial signals that activate specific responses in plants would help develop bioinoculants that allow for the establishment of crops in nutrient-poor soils or in the presence of phytopathogens.

## 5. Perspectives

Many works involving PGPRs have been published for over 50 years and different companies producing biologicals have emerged and, applied research using PGPRs has shown crop yield increments. However, the effects of PGPR applications on yield in different crops have been variable; in some cases increases in root length, foliage, and grain yield, were found when bacteria are used in combination with arbuscular mycorrhizal fungi. Knowledge about plant–microbe interactions using molecular and omics tools can help us improve these interactions and make the plants resilient to environmental changes due to global warming. Exploring the microbial diversity associated with crop wild relatives is also an alternative for detecting PGPRs with better phytostimulant characteristics. On the other hand, the sequencing of the PGPR genome allows for the identification of virulence factors to ensure that they do not pose a threat to human health or the environment as a result of their release.

The diversity and functionality of soil microbiota depend on the type of agricultural management, which is higher when crop rotation or intercropping systems are implemented. Tillage systems also have a decisive influence on the composition of microbial populations. The management of the soil microbiome is even more complex in the soil microbiome, the action of PGPRs is only a part of the agroecosystem in such a way that the introduction of foreign microorganisms to each site or plot does not always offer a positive behavior since the homeostasis of the soils tend to limit their establishment. Hence, a large number of local evaluations must be carried out.

The current challenge is to ensure crop success when using PGPRs to further encourage an increase in their usage, which must undergo rigorous evaluation methods to ensure product safety, viability, purity, and genetic stability. For the efficiency of PGPRs to be more significant, a series of factors must be considered, such as soil characteristics, type of plant, and agronomic practices, which determine the presence, dominance, or survival of specific bacteria populations (Figure 3).

## Figures and Tables

**Figure 1 plants-12-01668-f001:**
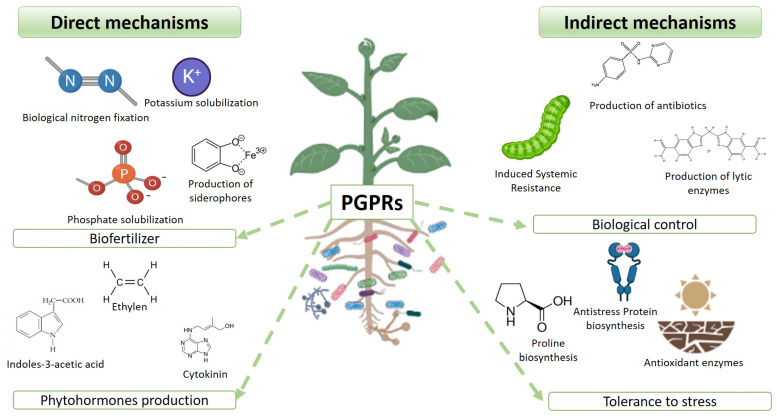
Direct and indirect mechanisms of PGPRs.

**Figure 2 plants-12-01668-f002:**
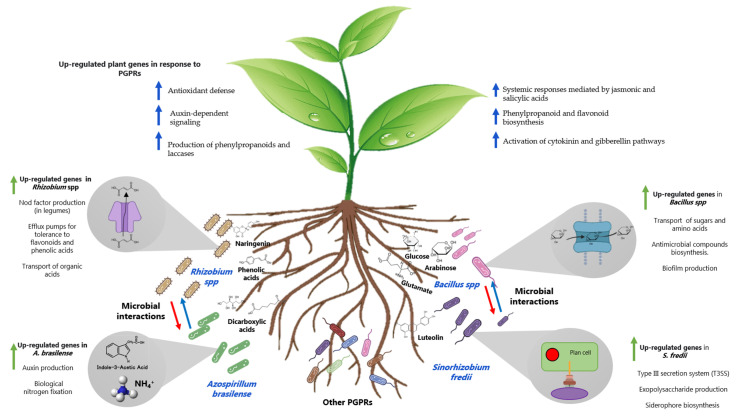
A transcriptomic view of plant–bacteria interaction.

**Figure 3 plants-12-01668-f003:**
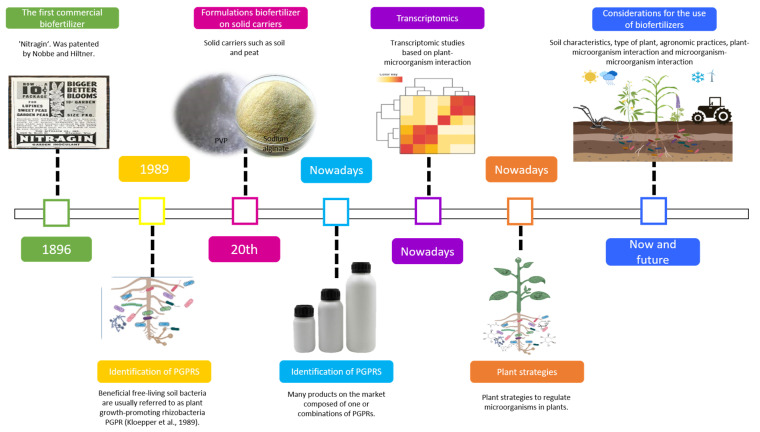
Past, present, and future of PGPRs as biofertilizers.

**Table 1 plants-12-01668-t001:** Microorganisms with different beneficial effects in different crops.

Bacterial Strains	Crop	Evaluation Conditions	Highlights	Reference
*Enterobacter hormaechei, Rhizobium* spp., *Pseudomonas fluorescence*, and AAULE51 (undetermined)	Pepper	Greenhouse using inoculated seeds.	Plants produced from inoculated seeds exhibited higher shoot and root lengths in addition to showing resistance to drought stress.	Admassie et al., 2022 [15]
*Bacillus subtilis* (MW644678, MW644686, MW644650, MW644649, MH845220, MZ488941, MZ488846), *Bacillus amyloliquefaciens* MW644651, *Bacillus safensis* MK212368, and *Bacillus halotolerans* MW644679.	Sugar beet	Under greenhouse conditions, using sugar beet seeds treated with each bacterium.	Antifungal activity against *Sclerotium rolfsii Sacc* and a reduction in the severity and incidence of root rot disease. Furthermore, increases in length of shoots and roots and plant fresh and dry weight were recorded.	Farhaoui et al., 2022 [16]
*Streptomyces corchorusii TKR8, Streptomyces corchorusii JAS2* and *Streptomyces misionensis TBS5*	Rice	Greenhouse conditions using inoculated seeds.	*Streptomyces*-treated plants exhibited improvement in rice plants’ growth and grain yield. Additionally, a reduction in the disease severity of bacterial panicle blight (BPB) was observed in treated plants.	Ngalimat et al., 2022 [17]
Plant Growth-Promoting Bacteria (PGPB) consortia	Oilseed rape	Pot experiment using Cd naturally polluted soil.	PGPB-based consortia promoted plant growth, increased Cd uptake of oilseed rape, Cd phytoextraction, and Cd removal from soil. Further, consortia increased microbial carbon, urease and sucrase activities, and the relative abundance of selected bacteria genera in soil.	Wang et al., 2022 [18]
*Enterobacter cloacae* and *Burkholderia cepacia*	Garlic	In vitro growth.	Both growth and physiological attributes of garlic were increased when their meristems were inoculated with the PGPB.	Costa Júnior et al., 2020 [19]
*Methylobacterium oryzae* MNL7 and *Paenibacillus polymyxa* MaAL70	Flooded paddy	100 g of field soil deposited into cork sealed beakers and filled up to 1.5 cm of water.	Grain yield and grain nutrient quality were improved by the co-inoculation; meanwhile, methane emission was reduced in comparison with uninoculated treatments.	Rani et al., 2021 [20]
*Paenibacillus taichungensis, Enterobacter* sp., *Rhizobium* sp., *Paenibacillus* sp., *Pseudomonas* sp., and *Paenibacillus pabuli*	Walker’s cattleya orchid	In vitro inoculation of seedlings obtained by micropropagation of *Cattleya walkeriana.*	Potential effect on improved nutrient acquisition and overall growth. Antioxidant enzyme activity and non-enzymatic antioxidants were increased.	Andrade et al., 2023 [21]
*Pseudomanas gessardi* EU LWNA-25 and *Erwinia rhapontici* EU-B1SP1	Amaranth	Controlled (pot) and natural (experimental farm) conditions.	Bacteria used as microbial consortia enhanced the growth of *Amaranthus* crops, expressed as the growth, grain, and yield.	Devi et al., 2022 [22]
*Acinetobacter calcoaceticus* P23, *Pseudomonas fulva* Ps6 and *Chryseobacterium* strains	Duckweed	Biomass production using a low nitrogen content and high salt food factory effluent (WW).	PGPRs promoted the growth of the crop under standard conditions but not when WW was used.	Khairina et al., 2020 [23]
*Azotobacter chroococum* and *A. vinelandii*	Eggplant	Root inoculation in plants exposed to different levels of drought stress.	Inoculated plants under drought stress exhibited higher dry matter production, leaf relative water, ions (K, Ca, and Mg), protein in roots, phenolic compounds, and proline.	Kiran et al., 2022 [24]

## Data Availability

Data sharing not applicable. No new data were created or analyzed in this study. Data sharing is not applicable to this article.

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
