# Peer review of "A Look at Plant-Growth-Promoting Bacteria"

_plants, 2023, doi:10.3390/plants12081668_

Round 1

Reviewer 1 Report

The paper "A look at plant growth-promoting bacteria" by Gómez-Godínez et al. is an interesting review that reports information on PGPMs, their role and the mechanism of action by which PGPMs exert their beneficial effects on host plants. The work is well written and reports interesting and quite up-to-date information.

To make their work more complete, I would suggest the authors add information on phyllospheric microbiomes and their role in improving plant performance under biothioc and abiotic factors.

Author Response

Thanks for your comments

We have done an English review.

We add in the document lines 155 to 175 information about microbiomes of the phyllosphere and their role in different conditions.

Changes can be viewed as track changes and all revisions.

Reviewer 2 Report

In the present paper, the use of bacteria to increase the yield of crops is analyzed. The beneficial effects that microorganisms offer to plants (such as biological nitrogen fixation, phosphorus solubilization, siderophores production, among others) are discussed.

Information regarding the strategies of plants to maintain microorganisms that may be beneficial to them is collected, such as the exudation of chemoattractant rhizodeposits for specific microorganisms and signaling pathways that regulate plant-bacteria interactions.

Finally, the contribution of transcriptomics is reviewed to try to elucidate plant-microorganism interactions.

General Aspects:

The general content of the article is adequate and complete.

The number of authors is adequate (in relation to the work carried out).

The language is correct. The wording is adequate and easy to understand.

The structure of the text is correct.

The bibliography is up-to-date and adequate.

I suggest that you reflect on the importance of verifying the harmlessness of the bacteria for further use in the formulation of biofertilizers. Current methods of massive sequencing can help to understand the virulence factors that the inoculum bacteria may carry and that may pose a threat to the environment as a result of their release.

Author Response

Thanks for your comments

We have done an English review.

You have a crucial point: The safety of a biofertilizer and the virulence factors of microorganisms in an inoculum. We have briefly reflected on this in lines 376 to 378 and 388 to 389.

However, we know that what you have mentioned is extremely important. It would be worth addressing a complete review of the safety of biofertilizers later on, focusing on virulence factors, plasticity, and genomic stability of microorganisms, as well as the prevalence and adaptation of biofertilizers to an autochthonous microbiota, not forgetting the different regions, environmental conditions and crops where they are applied.

Round 2

Reviewer 1 Report

Authors improved ms according to my comments.